# Effects of Fire Frequency Regimes on Flammability and Leaf Economics of Non-Graminoid Vegetation

**Arthur Lamounier Moura** [1,*] **, Daniel Negreiros** [2] **and Geraldo Wilson Fernandes** [2]

1 College of Forestry, Wildlife and Environment, Auburn University, Auburn, AL 36849, USA
2 Ecologia Evolutiva e Biodiversidade, Departamento de Genética, Ecologia & Evolução, Universidade Federal de Minas Gerais, Belo Horizonte 30161-970, MG, Brazil; dnapbr@ufmg.br (D.N.); gwilson@icb.ufmg.br (G.W.F.)
* Correspondence: arthurlamouniermoura@gmail.com

**Abstract:** Fire is an ecological factor that strongly influences plant communities and functional traits. Communities respond differently to fire, either decreasing or increasing in flammability and resource acquisition strategies. This study aimed to investigate the influence of fire over traits associated with flammability and the plant economic spectrum in a stressful and infertile mountainous grassland located in the Espinhaço mountain range in Brazil. Non-graminoid plant species were sampled in 60 5 m × 5 m plots distributed in three fire frequency categories. We measured several traits related to flammability—leaf dry matter content (LDMC), twig dry matter content, leaf area, bark thickness, branching architecture, plant height, leaf toughness (LT), and specific leaf area (SLA). Traits responded differently to the increase in fire frequency. For instance, the LDMC and LT were lower while the SLA was higher at high fire frequencies, indicating a trend towards reduced heat release and fire residence time. This shift resulted in the dominance of plants with a relatively more acquisitive strategy. This study brings evidence that traits respond coordinately towards a reduction of flammability with the increase in fire frequency and are strong indicators of the filtering role that fire plays as a disturbance on rupestrian grassland vegetation.

**Keywords:** fire ecology; fire regime; functional traits; community assembly; functional ecology; flammability; leaf economics spectrum; plant adaptations





## 1. Introduction

Tropical savannas are biomes formed under regular and natural fires between six and eight million years ago [1]. These biomes are characterized by the dominance of C4 grasses coexisting with stress-tolerant woody plants [2,3]. Many studies show that the expansion of C4 grasses was responsible for the increase in flammable vegetation. It was crucial for replacing some forest areas with savanna vegetation types we recognize today [1,4,5].

Among several other factors, such as moisture, soil nutrients, and temperature, the fire regime constitutes an important factor influencing savannas' plant community [3,6,7]. Fire regimes affect both individual species and functional groups, leading to variations in trait values and responses within a community [5,8,9]. In ecosystems that have recurrent burning cycles, fire drives physiological, reproductive, morphological, and ecological adaptations in plants [1,10,11]. The Brazilian Cerrado, for example, contains a set of families and species that exhibit fire tolerance and post-fire persistence traits, such as the post-fire flowering in the anemochoric sedge species *Bulbostylis paradoxa* and the leguminous shrub *Calliandra dysantha* [12,13], as well as bud-forming underground organs [14,15] and insulating corky bark [16]. Thus, fire acts as an important environmental filter that favors individuals with certain traits and excludes others [11,17,18]. Examples of favored traits and adaptations are resprouting ability [19], heat-induced germination [20–22], and post-fire mass flowering [11,23,24]. Another potential fire-related characteristic is the susceptibility

or resistance of a plant to produce flames when exposed to fire (that is, flammability) [25], one of the focuses of this study.

Flammability is a compound measure that integrates traits at different scales from the landscape level down to the organ and individual levels [25]. Examples of organ-level traits related to flammability are specific leaf area, dry matter content of leaves and twigs, bark thickness, leaf toughness, and leaf area. At the individual level, height and branching architecture are the ones most widely studied. Among these traits, specific leaf area and leaf area tend to be positively associated with high rates of combustion and low heat release, two dimensions of flammability [26–28]. On the other hand, traits such as leaf toughness, branching architecture, leaf dry matter content, and twig dry matter content are correlated with high heat release but low combustibility, resulting in intense flames with long residence time [25,29,30]. The traits bark thickness and height, in turn, have a negative relation with flammability as they confer protection against damage caused by flames [31,32].

The traits identified as key drivers of flammability are also associated with plants' resource-use strategies and leaf economics and describe how species acquire and invest resources [33]. For instance, the increase in leaf dry matter content is linked with an increase in heat release but is also correlated with the conservative end of the leaf economic spectrum, in which plants exhibit high density tissues and a longer leaf life span [34,35]. Conversely, a high specific leaf area represents an increase in combustibility but is a strong marker for fast-growing plants in the acquisitive side of the leaf economics spectrum, where plants exhibit higher photosynthetic rates and fast growth [34–36]. Therefore, since some traits are linked to both flammability and resource-use strategies, the effects of fire on the community's functional structure through flammability traits will also affect the character of vegetation resource-use strategies and leaf economics.

In fire-prone landscapes, Pausas [25] describes three major plant strategies related to fire that are associated with organ and individual traits. The first is called the non-flammable strategy, commonly comprised of species whose traits result in low flammability (e.g., increased height, reduced branching). These traits enable them to survive even in systems dominated by flammable species, such as grasslands and savannas [37]. The second strategy is called fast-flammable and comprises species that feature traits that cause them to burn quickly (e.g., high specific leaf area, high leaf area) but with overall low heat release and low fire residence time. These characteristics allow them to survive and preserve organs and tissues undamaged [38]. The third strategy comprises the hot-flammable species, whose traits produce intense flames with high heat release (e.g., high leaf dry matter content, twig dry matter content, and leaf toughness). These traits often do not allow the individual to survive due to the fire intensity but are associated with massive post-fire recruitment in low-competition post-fire scenarios. All these strategies are key for the survival and persistence of plants in fire-prone landscapes.

Currently, fire's influence on plants is being increasingly modified by changes in fire regimes caused mainly by land-use change and global warming [39,40]. In this context, alterations in the natural regime could result in changes in dominance ranking between autochthonous species as well as the exclusion or replacement of species that do not tolerate different patterns of fire, allowing the introduction of other species and strategies that did not previously occur in the habitat [41,42]. This scenario could cause profound modifications in the vegetation functional structure, either with more frequent, more intense, or more extensive fires [9,43,44]. Because of that, landscapes that are undergoing modifications in their fire regime create opportunities for research aiming to detect how fire shapes plant strategies and the vegetation functional structure. These modifications in the fire regime are currently happening to the rupestrian grasslands in the few past decades, a vegetation complex within the Brazilian tropical savanna Cerrado, where abrupt changes in the fire frequency took place as a result of human activities, such as cattle raising, biological invasions, and agriculture, and burning has become more frequent in some areas [17,41,45,46].

The aim of our study is to answer how alterations in fire frequency regimes within a fire-prone ecosystem influence the traits associated with flammability and resource-use strategies in plants. To achieve this goal, we hypothesize that the increase in fire frequency will select for traits that reduce fire intensity, such as fast-flammable traits (e.g., decreased leaf dry matter content and twig dry matter content and increased specific leaf area) or those that prevent burning (e.g., increased height and decreased branching). These traits would enhance plant survival and reduce tissue damage under repeated fires, which are important mechanisms for protection in fire-prone landscapes. Consequently, increased fire frequency would favor dominating species that exhibit a less conservative strategy according to the plant economic spectrum and resource-use strategy.

## 2. Materials and Methods

### 2.1. Study Site

The study was conducted in two different areas in Serra do Cipó—Reserva Vellozia and Lapinha da Serra—both located in the District of Santana do Riacho in the southern portion of the Espinhaço mountain range in Minas Gerais, Brazil (Figure 1). Vegetation sampling was carried out between October and December 2018. The dominant ecosystem in these areas is the rupestrian grassland (campo rupestre in Portuguese), an extra zonal, species-rich vegetation complex found in Brazil, frequently occurring in altitudes above 900 m [47]. Occurring within the Cerrado biome, the rupestrian grassland is classified as an OCBIL (old, climatically buffered, infertile landscape) whose soil is characterized by its seasonal water scarcity and chronically extremely low-nutrient levels [48]. This vegetation is composed of grassy, herbaceous, and shrub layers associated with litolithic soils, disposed in a mosaic with rocky outcrops of quartzite, sandstone, or ironstone [48–51].

A critical characteristic of the rupestrian grasslands is that, similarly to other Cerrado plant formations, its vegetation is directly associated with fire, with evidence that ignitions occurred naturally in these landscapes even before human occupation [17]. Currently, the rupestrian grasslands are experiencing rapid changes in fire regimes due to various human activities, including cattle raising, biological invasions, and agriculture. As a result, some areas are experiencing more frequent and intense burning in recent decades [17,41,45,46]. As these anthropogenic processes continue to intensify and serve as the primary source of ignitions, it becomes crucial to utilize remote sensing studies that can reconstruct fire history both spatially and temporally in fire-prone landscapes [45]. Such studies provide invaluable support for understanding the influence of fire regime changes on vegetation dynamics.

### 2.2. Fire Regime Classification

To assess the fire regimes of the study area, we used the temporal sequence of fires made by Alvarado [45], in which the incidence of fire was mapped using Landsat satellite images from 1984 to 2014. This comprehensive study enabled us to spatially and temporally analyze the areas that burned and the frequency at which they burned, providing crucial insights into the current state of the fire regime in the area. Based on the distribution and the amplitude of fire return intervals found in the study area by Alvarado [45], we partitioned the area into three equal parts, each representing a specific range of fire return intervals: low (more than four years of fire return interval), medium (between two and three years of fire return interval), and high (less than two years of fire return interval) along with 30 years of fire history (Figure 1; Supplementary Table S1).

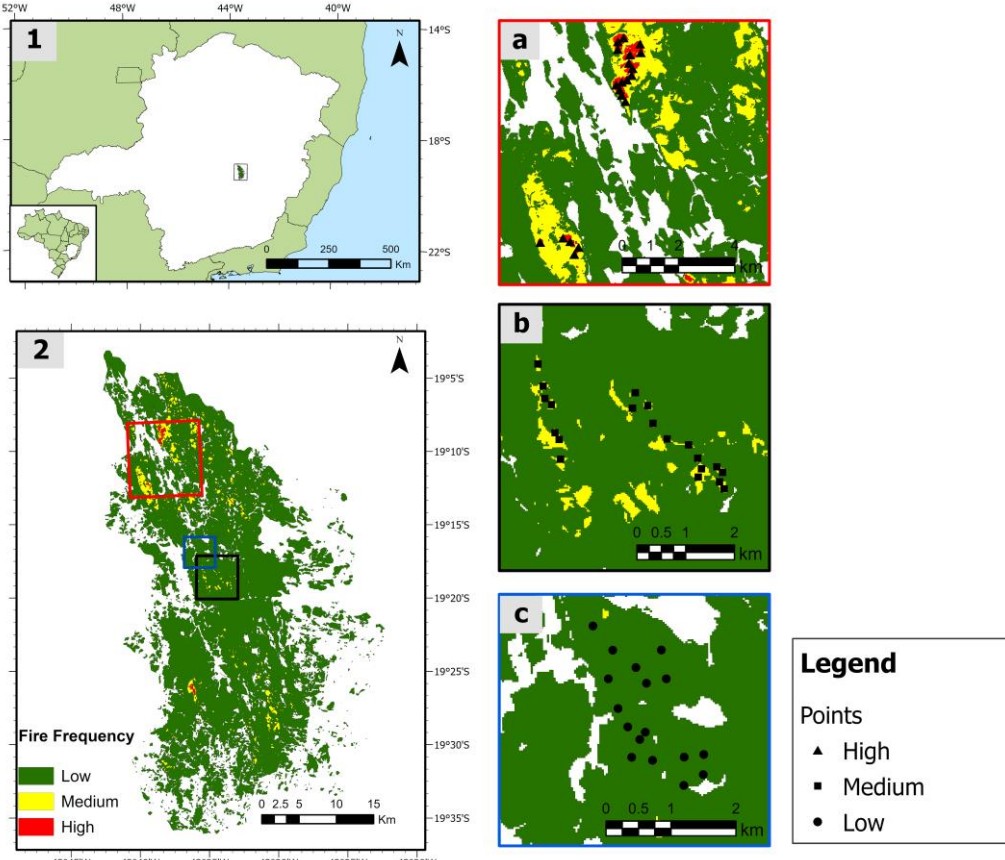

**Figure 1.** Map of the study area in the District of Santana do Riacho, Minas Gerais, Brazil, where we sampled functional traits in the rupestrian grassland non-graminoid vegetation. Box **1** provides an overview of the study area within the state of Minas Gerais, Brazil, while Box **2** includes a more detailed representation of the study area and its fire frequency regimes. The green color in Box **2** represents areas with low fire frequency (more than four years of fire return interval), the yellow color represents areas under medium frequency (between two and three years of fire return interval), and, in red, the areas under high fire frequency (less than two years of fire return interval). This map was modified from the fire history reconstruction made by Alvarado et al., (2017) [45]. The colored squares in Box **2** are shown in detail in boxes (**a**–**c**), where the sampling points are represented in geometric shapes (●, low-frequency; ■, medium-frequency; and ▲, high fire frequency).

### 2.3. Sampling

After categorizing the fire regimes, we selected the sites where the sampling took place. To avoid bias, we randomly generated 20 sampling points within the perimeters of each fire regime category (total of 60 points) at approximately the same altitude and the same vegetation type, the rupestrian grassland. Additionally, we excluded areas that could be influenced by potential anthropogenic confounding factors, such as grazing, trampling, and other agricultural activities known to affect vegetation composition. To avoid autocorrelation between the plant communities due to dispersion and colonization, the points were distant by at least 200 m, preferably with geographical barriers between them, such as rocky outcrops and slopes. In each of these points, we established a quadrat of 5 m × 5 m in which the non-graminoid stratum was observed, thus excluding Poaceae (grasses) and Cyperaceae (sedges) individuals. The species that together corresponded to 70–80% of plot abundance—dominant species—were chosen for sampling and identification. To identify the dominant species, we ordered the plants based on the percentage of total abundance and the set of species that together met the required percentage of 70–80% threshold was selected for sampling and trait measurement. We randomly selected

8–10 mature, healthy-looking individuals (no signs of herbivory or pathogens) of each species in the sampling points to obtain the functional trait measurements, including twigs, leaves, and living plant measurements (e.g., height, branching architecture). The sampling was based on the protocols of Cornelissen [32] and Pérez-Harguindeguy [52]. We selected non-graminoids for this study because they are a diverse group of plants that exhibit a wide range of flammability traits relative to graminoids [25]. This allowed us to test the effects of fire frequency on a variety of flammability traits, both at the organ (e.g., leaves, twigs) and at the individual level, and improve the comparisons between fire frequency categories.

### 2.4. Functional Traits

The plant functional traits we selected to measure were associated with flammability, the set of characteristics related to the susceptibility or resistance of a plant to produce flames when exposed to fire. From this group, we assessed traits at the organ level: leaf dry matter content (LDMC), twig dry matter content (TDMC), leaf area (LA), specific leaf area (SLA), and leaf toughness (LT) as well as traits at the individual level: height (H), branching architecture (BR), and bark thickness (BT) (Figure 2). For foliar traits and twigs, we sampled 2 of each organ per individual. After sampling, leaves and twigs were labeled and stored in a cooler at approximately 4 °C for laboratory processing within 24 h. Both fresh and dry weights were obtained with an analytic scale with precision of 0.1 mg. For dry weights, leaves and twigs were oven dried at 60 °C for 72 h. We obtained LA measurements using a desktop scanner, and BT measurements were taken with digital calipers to the nearest 0.01 mm. At the individual level, H was obtained with a graded meter tape to the nearest 0.1 cm from ground level to the highest point where photosynthetic tissues were present. The BR was taken by counting the number of ramification orders in two stems per individual. Specifically for LT, we obtained the results through the use of a digital penetrometer (Chatillon® model DFE-010, Largo, FL, USA) coupled to a cone-shaped tip (model SPK-FMG-009A) according to Silva and Batalha [53].

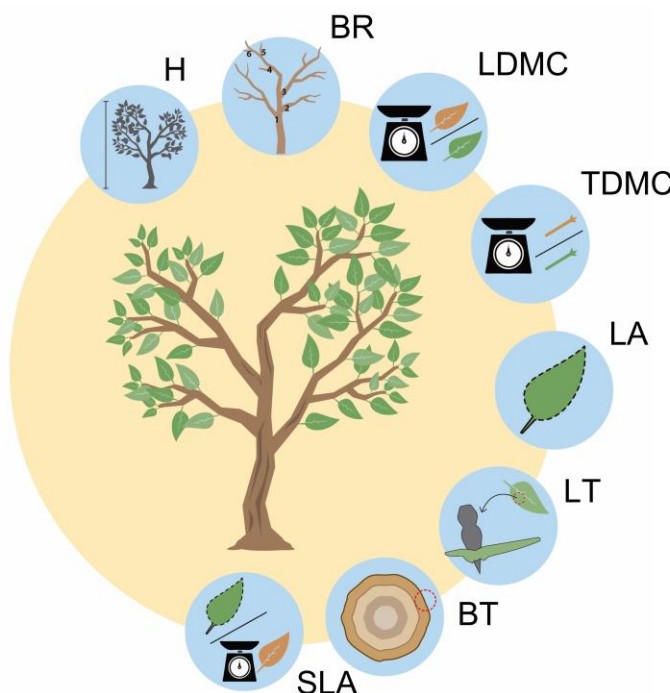

**Figure 2.** Set of traits measured in the most common rupestrian grasslands non-graminoid species, which together accounted for 70–80% of the total abundance in the sampling points of three fire frequency regimes (low, medium, and high). The traits are abbreviated as H (height), BR (branching architecture, LDMC (leaf dry matter content), TDMC (twig dry matter content), LA (leaf area), LT (leaf toughness), BT (bark thickness), and SLA (specific leaf area).

### 3. Data Analysis

Before running the analyses, we calculated two means, the mean of trait replicates measured in the same individual, and the mean of traits between individuals of the same species in a given sampling site. Consequently, each species in a site showed one value for the measured trait. To test for differences among trait values in contrasting fire frequency regimes, we ran linear mixed effect models (LMM) and linear fixed effect models (LM) in the software R Version 3.5.2 (R Core Team 2018) using the plant's functional traits as response variables and the fire frequency regimes as the independent variable. For the LMM, we used the sampling points nested within the fire frequency regimes as the random effect variable to account for any between-sampling point variation. We then used F-tests to compare both LMM and LM models and identify the model that better fits the data. We checked both residual normality and homogeneity of variance. To better understand the patterns between the means of traits in each fire regime, we also performed post hoc Tukey's HSD pairwise comparisons.

### 4. Results

We sampled a total of 559 individuals of 38 species from 14 families in all fire frequency regimes. Asteraceae, Melastomataceae, and Velloziaceae were the most represented families, with eight, six, and five species, respectively (Supplementary Table S2). Fifteen species were sampled in low and high-frequency sites, and sixteen species were sampled in medium-frequency sites. Of the 38 species we studied, 4 were found in two of the three fire frequency regimes, and only 1 species, *Vellozia variabilis*, was found in all three regimes.

The LM showed to be the best fit model for all functional traits we measured, which led us to select it as our statistical model hereafter. We found that there was a statistically significant difference in the mean LDMC between at least two groups. Tukey's HSD test for multiple comparisons found that the mean value of the LDMC was significantly different between high-frequency regimes and low-frequency regimes ($p = 0.0067$) (Figure 3a). The specific leaf area, in turn, was significantly higher under the high-frequency regime when compared to medium- and low-frequency fire regimes (Figure 3b). On the other hand, the LT was significantly lower under the high-frequency regime compared to the low-frequency, whereas in the medium-frequency, LT was in an intermediate state between the two (Figure 3c). Lastly, Tukey's HSD test showed that the BR showed a different pattern, in which plants under the high ($p = 0.001$) and low fire frequency ($p = 0.002$) regimes featured significantly higher numbers of branches, contrasting with the medium-frequency regime that showed a reduced number of branches (Figure 3d). All the statistical details for the comparison between traits and fire regimes are reported in Table 1. The traits TDMC, LA, H, and BT did not show any statistical difference between fire frequency regimes ($p > 0.05$).

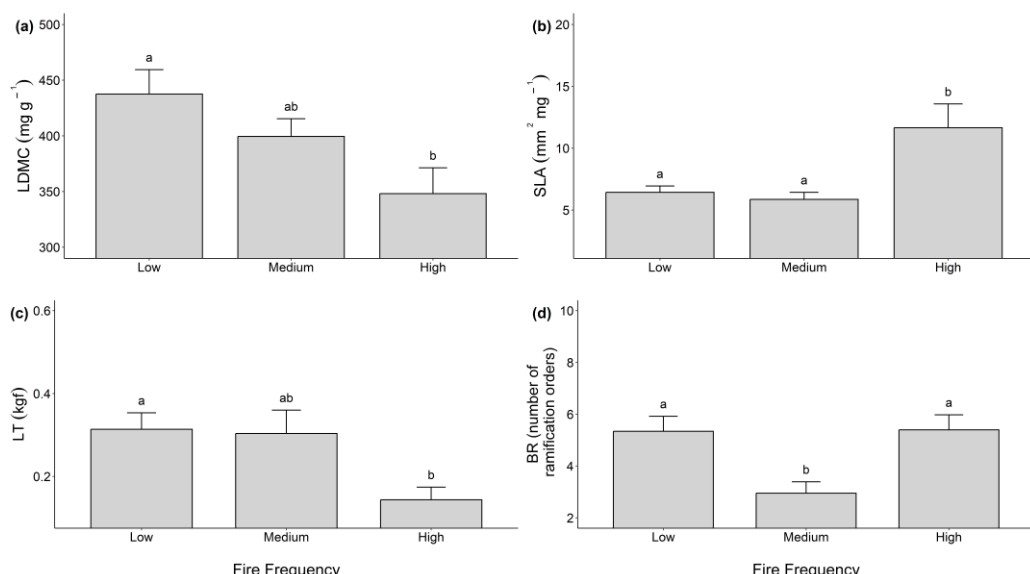

**Figure 3.** Bar plots displaying the mean and standard error of functional traits in each fire frequency regime (low, medium, and high). (**a**) leaf dry matter content, (**b**) specific leaf area, (**c**) leaf toughness, (**d**) branching architecture. Fire frequency regimes with different letters indicate statistically significant differences at a probability level of <0.05, based on Tukey's HSD test.

**Table 1.** Summary of linear models and Tukey's HSD tests for multiple comparisons contrasting the functional traits of rupestrian grassland non-graminoid vegetation between the three fire frequency regimes (low, medium, and high) with mean and ±95% confidence interval. Fire frequency regimes with different letters indicate statistically significant differences ($p < 0.05$) between groups based on Tukey's HSD test. The last column shows the F-test model comparisons between linear mixed effect (LMM) model and linear fixed effect model (LM), where a $p$-value > 0.05 means that the linear fixed effect model, without the random variable, better fits the data. Abbreviations are as follows: Resid. Df (Residual degrees of freedom), Resid. Dev (Residual deviance), LDMC (leaf dry matter content), SLA (specific leaf area), LT (leaf toughness), BR (branching architecture), H (height), TDMC (twig dry matter content), LA (leaf area), and BT (bark thickness).

| Trait | Fire Regime | | | Deviance | Resid. Df | Resid. Dev | F | $R^2$ | $p$-Value [1] | F-Test $p$-Value (LMM vs. LM) |
| --- | --- | --- | --- | --- | --- | --- | --- | --- | --- | --- |
| | Low | Medium | High | | | | | | | |
| LDMC (mg g$^{-1}$) | 437.57 ± 41.98 a | 399.45 ± 52.78 ab | 348.14 ± 49.66 b | 81,323 | 68 | 541920 | 5.1022 | 0.131 | **0.0086** | 0.254 |
| SLA (mm$^2$ mg$^{-1}$) | 6.44 ± 2.44 a | 5.86 ± 3.06 b | 11.66 ± 3.29 b | 477.09 | 68 | 1832.7 | 8.8512 | 0.207 | **0.0003** | 0.341 |
| LT (kgf) | 0.313 ± 0.13 a | 0.303 ± 0.10 ab | 0.143 ± 0.14 b | 0.36524 | 58 | 2.4403 | 4.3404 | 0.13 | **0.0175** | 0.7892 |
| BR (# order of ramifications) | 5.34 ± 1.10 a | 2.95 ± 1.38 b | 5.39 ± 1.48 a | 102.28 | 68 | 373.56 | 9.3097 | 0.215 | **0.0002** | 0.4088 |
| H (m) | 1.14 ± 0.23 a | 0.79 ± 0.29 a | 0.95 ± 0.31 a | 1.43 | 68 | 16.855 | 2.8847 | 0.078 | 0.0627 | 0.138 |
| TDMC (mg g$^{-1}$) | 450.44 ± 52.33 a | 469.23 ± 67.57 a | 398.15 ± 71.03 a | 55,424 | 56 | 611651 | 2.5372 | 0.08 | 0.0881 | 0.9545 |
| LA (mm$^2$) | 1435.61 ± 2492.42 a | 1252.39 ± 3133.57 a | 3250.32 ± 3360.79 a | 408,041 | 67 | $2.00 \times 10^8$ | 0.0523 | 0.029 | 0.9491 | 0.6831 |
| BT (mm) | 3.44 ± 3.36 a | 3.89 ± 4.28 a | 4.23 ± 4.58 a | 6.1287 | 65 | 3320.5 | 0.06 | 0.002 | 0.9418 | 0.9887 |

[1] $p$-values in bold indicate significant differences among fire regimes.

## 5. Discussion

Some key functional traits measured in this study were consistently associated with the fire regime in the study area and responded accordingly to the increase or decrease in the fire frequency: leaf dry matter content, specific leaf area, leaf toughness, and branching architecture. These traits are strongly related to both flammability and leaf economics. Several studies in different systems have shown that disturbance by fire is an effective environmental filter able to determine traits and patterns present within plant communities [5,9,11,42,54]. However, although the goal of this study was to reveal the extent of fire frequency regime's influence on flammability traits, not all traits responded equally to variations in fire frequency. This suggests that the predictive power of fire frequency is stronger for some traits and weaker for others. In fact, functional traits' adaptive value is associated with a combination of environmental factors that the individuals experience, as fire is an important but not exclusive factor [55].

Leaf dry matter content decreased under high-frequency sites compared to low-frequency sites, with medium-frequency sites appearing to be in an intermediate state between low or high, not showing statistical differences between them. This indicates that increased frequencies of fire may favor species with reduced leaf dry matter content, a trait reported to be positively related to heat release and fire residence time [25,56–58]. This relationship was found in Abedi [59] and is aligned with the hypothesis that increased fire recurrence leads to a dominance of traits that would result in reduced fire intensity. That, in turn, potentially provides a higher survival during the passage of fire, especially in a fire-prone landscape where grasses increase the flammability of the community [25,60]. We suggest that the resilience of a set of species and its dominance in high fire frequencies might be related to the reduction of heat release and fire residence time in the leaves, which is linked with reduced tissue damage [25,29]. As a consequence of the decreased leaf dry matter content in higher frequency regimes, a relatively more acquisitive strategy is favored, since leaf dry matter content is negatively correlated with potential growth rate and leaf life span [32,61]. Future studies could improve this conclusion by testing it in long-term prescribed fire experiments and observing the impacts on plant survival and growth.

Specific leaf area, in turn, was greater in high fire frequency sites when compared to medium- and low-frequency sites. In terms of flammability, this pattern is aligned with the fast-flammable strategy hypothesis, in which plants that exhibit high specific leaf area have a higher survival and preserve organs and tissues undamaged due to low heat release and low fire residence time during the burn [27,28]. This is also consistent with the results we found for leaf dry matter content, which may be the most prevalent strategy in the rupestrian grasslands under increased fire frequency. Additionally, this result also means that high fire frequency favored more acquisitive and fast-growing species, with leaves featuring relatively higher light interception, lower life span, and reduced antiherbivore defenses [32,52]. This corroborates the hypothesis that high-frequency fire regimes select plants with a relatively faster growth rate as they are more likely to persist in areas disturbed by fire through rapid regrowth and regeneration after disturbance [42] and is consistent with findings in experimental and simulated studies that found an association between high specific leaf area values and high levels of disturbance and functional clustering under high fire frequency [62]. Specific leaf area is not only important for plant growth, but also to the fast regrowth of sprouts, a commonly found strategy in the rupestrian grassland and fire-prone ecosystems [23,63].

Additionally, plants that inhabited high fire frequency sites showed decreased leaf toughness compared to those in low fire frequency areas. Similar to leaf dry matter content, leaf toughness is linked with leaf tissue density and lower values for this trait are associated with low heat release and low residence time, therefore reducing intense, damaging flames [25,32,64]. Again, medium-frequency sites appear to be in an intermediate state and do not show statistical differences between low or high. This trend also indicates that increased frequencies of fire may favor lower heat release and lower fire residence time, aligned with the fast-flammable strategy. Additionally, higher values of leaf toughness are

associated with the conservative end of the economic spectrum [65]. Thus, this pattern is consistent in this study, meaning that the occurrence of reduced leaf toughness and leaf dry matter content and increased specific leaf area is indicative of relatively more acquisitive strategy. It means that under higher fire frequencies, there is habitat filtering favoring plants with reduced leaf toughness, relatively higher specific leaf area, decreased leaf life span, and dry matter content.

Branching architecture, which is positively correlated to heat release [29,66], displayed a unique pattern. In areas with low and high fire frequencies, there were a similar number of branches, whereas medium-frequency areas had fewer branches. Investing in ramification can increase the plant's growth rate, carbon gain, and ability to use light efficiently [67]. However, having more branches also increases the plant's flammability and the likelihood of a fire spreading to the aerial parts of plants [32,67]. From the perspective of plants under high fire frequency, the trait is advantageous to shoot growth response after recurrent fires [68]. In addition, considering the recurrence of fire disturbance in these areas, featuring a highly branched architecture could enable fast growth to escape fires or even the acceleration of the vegetative life cycle before another fire occurs [35,69]—surpassing the risks related to flammability. Medium-frequency sites appeared as intermediate states for leaf dry matter content, specific leaf area, and leaf toughness, but for branching architecture, the results are consistent with our hypothesis and point to a reduction in the branch ramification and flammability. While high-frequency sites were shown to be associated with a trade-off between fast growth and flammability, medium-frequency sites seem to be linked with higher survival, a reduction in shoot flammability, and protection of aerial parts [66,70]. Although medium-frequency sites are not in the extreme end of fire recurrency in the rupestrian grasslands, fire return is still filtering for species with less flammable branching architecture. On the other hand, for the plants sampled in the low fire frequency areas, exhibiting a highly branched architecture is also advantageous in terms of productivity since a higher number of branches makes a higher amount of productive tissues possible. However, these plants do not experience the risks of flammability that the plants found under high-frequency regimes do because fire is not a recurrent disturbance in these environments. Thus, the observed plants are less affected by the risks of intense and damaging fires. Moreover, the time interval between fires is considerably greater than in any other regime, resulting in a more durable aerial part, possibly from older individuals that were able to branch proportionally.

Together, these traits contribute to the reduction of tissue damage, low heat release, and a fast-flammable strategy, consequently resulting in the dominance of plants with a relatively faster growth rate, which are more likely to persist in areas disturbed by fire through rapid regrowth and regeneration after disturbance. However, it is important to emphasize that, although there is an increase in the dominance of strategies relatively more acquisitive, the rupestrian grasslands still constrain the species to an overall conservative strategy due to its extremely infertile soils and stressful environment [48,71]. Additionally, it is crucial to note that the functional shifts and trends observed in our study should not be understood as globally applicable to all scales, vegetation types, or examined functional traits. Distinct patterns are found in a different set of functional traits [44], distinct scales of study [72], and different geographic regions [72,73]. Therefore, it is important to consider specific contexts and factors when interpreting the effects of fire frequencies on vegetation. Future studies analyzing the vegetation responses across different regions in the globe will contribute to advancing our understanding of the functional trends and general responses of vegetation to increasing fire frequencies. However, our study provides evidence that ongoing disturbances to this highly threatened biodiversity hotspot can alter the functional structure of the vegetation, thereby impacting its biodiversity. Our findings are consistent with predictions regarding the consequences of land-use changes and anthropogenic threats to the rupestrian grasslands [74–76].

Regarding the traits that did not differ among fire frequency regimes, we can highlight that bark thickness showed no correlation with fire frequencies. It can be indicative that

this trait is not the main source of mortality avoidance in the rupestrian grassland as it is in other ecosystems [56,63,77]. More studies exploring the role of bark in the rupestrian grassland species could help the interpretation of this pattern, but this may be evidence that in pyrophytic vegetations, plants can develop different strategies of survival, such as resprout, a widely distributed strategy in the rupestrian grassland ecosystem [23,78,79].

## 6. Conclusions

Our study showed that the main functional traits responded coordinately along the fire frequency gradient, mainly pointing towards a fast-flammable strategy and shifting to the acquisitive side of the plant economics spectrum. It is important to highlight the potential of fire as a predictor of functional trait distribution and responses in plant species assemblages in the rupestrian grassland ecosystem. Hence, we evidenced the potential of functional traits to entangle flammability and leaf economics. It is of major importance to expand the understanding of fire dynamics in the rupestrian grassland ecosystem to identify other influences that this disturbance produces in plant communities and other ecological processes. Phylogenetic studies, for instance, could give support to our results, revealing evolutionary patterns underlying the analyzed functional traits present in each fire regime.

**Supplementary Materials:** The following supporting information can be downloaded at: https://www.mdpi.com/article/10.3390/fire6070265/s1, Table S1: List of families and species sampled in the rupestrian grassland non-graminoid vegetation, and the corresponding fire frequency regimes in which they were sampled. The fire frequency values indicate the number of plots where species were found. The growth form and presence/absence of underground storage organ information is based on Flora e Funga do Brasil. Available online: http://floradobrasil.jbrj.gov.br (accessed on 6 June 2023). All species have a perennial life cycle. NA—Not available; Table S2. List of plots categorized by fire frequency regime (low, medium, high), indicating the time elapsed since the last fire at the time of vegetation sampling and the year of the last fire event.

**Author Contributions:** A.L.M.: Conceptualization, Data curation, Formal analysis, Investigation, Methodology, Project administration, Software, Validation, Visualization, Writing—original draft. D.N.: Data curation, Formal analysis, Validation, Writing—review & editing. G.W.F.: Funding acquisition, Project administration, Resources, Supervision, Validation, Writing—review & editing. All authors have read and agreed to the published version of the manuscript.

**Funding:** This research was funded by Minas Gerais Research Funding Foundation (Fapemig) grant number APQ 01477-14 and by the National Council for Scientific and Technological Development (CNPq) through the Long-Term Ecological Research "PELD Campos Rupestres da Serra do Cipó" grant number 442694/2020-2. GWF received a Productivity Research Fellowship from CNPq grant number 316258/2021-0.

**Institutional Review Board Statement:** Not applicable.

**Informed Consent Statement:** Not applicable.

**Data Availability Statement:** Not applicable.

**Acknowledgments:** We thank FAO Silveira, EKL Batista and M Barbosa for rich comments on the manuscript and ST Alvarado for her invaluable time in providing us with the fire frequency maps and information about the time since the last fire in our sampling locations. We also thank NPU Barbosa for its contributions to conceptualizing this study . . . We thank BS Ferreira, WK Siqueira, LA Garcia, J Santiago, and DC Pinheiro for helping with lab and fieldwork; WK Siqueira also helped with the statistical analysis and BS Ferreira with plant identification. We are also grateful to Reserva Vellozia for providing logistical support to develop this study.

**Conflicts of Interest:** The authors declare that they have no competing financial or personal interests that could have influenced the development of this study.

## Abbreviations

| Abbreviation | Definition | Unit |
|---|---|---|
| BR | Branching architecture | # order of ramifications |
| BT | Bark Thickness | mm |
| H | Height | m |
| LA | Leaf Area | $mm^2$ |
| LDMC | Leaf Dry Matter Content | $mg\,g^{-1}$ |
| LT | Leaf Toughness | kgf |
| SLA | Specific Leaf Area | $mm^2\,mg^{-1}$ |
| TDMC | Twig Dry Matter Content | $mg\,g^{-1}$ |

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
