# Peer review of "Effects of Fire Frequency Regimes on Flammability and Leaf Economics of Non-Graminoid Vegetation"

_fire, doi:10.3390/fire6070265_

Round 1
Reviewer 1 Report (Previous Reviewer 1)
The revised version has shown significant improvement by incorporating all the suggestions provided in the initial version.
However, I have a specific question pertaining to the discussion: the extent to which the observed response towards a more acquisitive resource-use strategy in environments with high fire frequencies can be generalized (L 260-264). Notably, there are studies conducted in other South American environments that have reported an opposing response pattern (Carbone & Aguilar, 2016, doi:10.1111/aec.12364; Loiola et al., 2020) or a neutral one (Santacruz-Garcia et al., 2019, doi:10.1111/aec.12815) concerning functional traits and flammability. A brief discussion on the potential reasons underlying these divergent outcomes across regions would enhance the generalizability of the findings and highlight the need for future cross-regional analyses to elucidate the general trends pertaining to functional traits, flammability, and fire.
Author Response
Please see the attachment.

Reviewer 2 Report (Previous Reviewer 3)
This study investigated the plant traits in association with their flammability and leaf economics in a mountainous grassland in Brazil. Non-graminoid plant species were selected, and several plant features and parameters were measured, including leaf dry matter content, twig dry matter content, leaf area, bark thickness, branching, plant height and leaf toughness and specific leaf area. It was found that plant traits responded differently to the increase in fire frequency. This study somehow brings evidence that plant features respond coordinately towards a reduction in flammability with the increased fire frequency.
Although corrections have been made in the revised manuscript, obvious issues are still there, especially in the arrangement of the technical contents. All the tabulated information and figure data must be inserted in text at their right places rather than separated as Appendix or supplementary materials. Meanwhile, the first two paragraphs in the section of Discussion should be merged in the Results section. The last paragraph in the discussion section should be presented independently to form the section of Conclusions.
Given that lots of abbreviations are used in the text, a list of abbreviations should also be provided as an integrated part of the text. This journal has specific format in the show of the bibliographical information of the references; as a result, the format of the listed references should be adjusted accordingly. After making these suggested changes, the manuscript should look better and more readable as a technical paper.
Acceptable.
Author Response
Please see the attachment.

Reviewer 3 Report (New Reviewer)
General Comments:
Interesting manuscript regarding plant species traits related to flammability in fire-prone Brazilian rupestrian grasslands. While the study seems to have been rigorously conducted much information about the plants, fire regimes, methods and natural conditions of these system remains unclear and needs to be specified, see below for these general comments.
Firstly, the selection of only some plants (e.g., the exclusion of graminids) and the selection of ’species that together corresponded to 70 – 80% of plot abundance..’, makes it unclear how these plant communities are overall structured. It removes important relationships of species richness and individual plant frequency, these would have been helpful for the reader to understand not only the overall composition, but evenness as well. This also contributes to the uncertainty of having a few species, that can be present within each plot by random chance, to dominate the analyses without much context.
Secondly, and related to the first point, there is a lack of relevant information about the plants: flammable compounds, life strategies, life-forms, seed-strategies, presence of storage organs etc., as these are likely important in determining the relationship to the fire-regimes.
Thirdly, the selection of the fire frequency categories is not clear and will need to be clearly specified and motivated. This is especially relevant for the ‘low’ category, as the fire return interval for some plots might as well be 20 years.
Fourthly, and related to the third point, how is the fire regime categorized? Is it natural or anthropogenic? How long has it been ongoing in these systems? Does it differ between areas? Does the fire-return interval since 1984 vary temporally between the plots within each category? What has caused it to change from the ‘earlier’ condition, and has this additional change brought on further factors into the system, e.g., grazing, fertilization from animals, trampling or other agriculturally related aspects? What is the plant succession process like?
Fifthly, there is no information about plot variation within each fire frequency category, how large is it and does it vary based on other environmental gradients? Could the results be due to random chance of plant establishment? Perhaps the result is further influenced by the fact that most plots seem to have been burnt around the same time (Supplementary Table 2), potentially influencing the result more than the general fire return-interval.
Sixthly, it would have helped a lot if the plot selection was randomized within areas, as all of the plots within each category is spatially close while each category is separated, rather than intermingled. This needs to be addressed and potential consequences of this, as the species distribution (and thereby the trait variation) could be caused by geographical or environmental similarity. Furthermore, is there any information regarding trait variation in isolated unburnt control plots that can strengthen the narrative?
In conclusion, there are a lot of questions and missing information that needs to be incorporated into the manuscript for proper interpretation of the findings. When dealing with such a small sampling size of plants in a spatially and temporally biased plot design much more information is needed to draw large general conclusions of the plant traits.
Specific Comments:
1. Lines 143-144. It is not clear what the 70 – 80 % interval refers to, what was the cut-off for including individual species in the trait measuring?
Round 2
Reviewer 2 Report (Previous Reviewer 3)
This study investigated the plant traits in association with their flammability and leaf economics in a mountainous grassland in Brazil. Non-graminoid plant species were selected, and several plant features and parameters were measured, including leaf dry matter content, twig dry matter content, leaf area, bark thickness, branching, plant height and leaf toughness and specific leaf area. It was found that plant traits responded differently to the increase in fire frequency. This study somehow brings evidence that plant features respond coordinately towards a reduction in flammability with the increased fire frequency.
Significant corrections have been made in the secondly revised manuscript, and it sounds that everything is there except a few small issues. It is inappropriate to include any citations in the conclusion section. So, essential changes should be made in the Conclusion part and the reference list. In addition, all the mathematical symbols should be placed in italicized throughout the paper. Please make appropriate changes and submit the final version of your paper for publication.
It's ok.
Author Response
Please see the attachment.

Reviewer 3 Report (New Reviewer)
General Comments:
The authors have tried to incorporate some of the reviewers’ concerns, yet many comments remain unaddressed.
Comment 1 (Firstly..) and the specific comment: It is still not clear what the method was of selecting species based on ’70 – 80%’. How was this observation of abundance per species selected? The reviewer understands that common species was selected, but how? Depending on how frequencies add up it is easy to select one species with 5% abundance and one with 65% abundance to make it up to total 70%, but perhaps one species had the final 30% abundance, but the species with 5% was selected or seen first, hence the method is not clear. The comment emphasis is on ‘together corresponded’, how?
Comment 2 (Secondly..): Without intimate knowledge of the study system and the species therein it is impossible to know what type of species were included in the study without looking up each individually. This is why further species information is needed. The information is not necessarily needed in the analysis itself, as the authors state, but within supplementary table S1. It is relevant for the reader to know if a species is a perennial shrub, a low herb or a tree seedling, and if the plant has underground storage organs that results in quick recovery or is completely dependent on the seedbank or nearby dispersion.
Comment 3 (Thirdly..): The expansion of the section did not address the reviewer question. The comment was regarding why 4 years or more was selected as ‘low’, rather than 5 years or more, or 10 years or more. If the same exact low, medium and high intervals were taken from Alvadaro et al 2017, then this should be stated. If the intervals were not taken directly from this paper, then the authors need to state why they regarded ‘4 years and more’ as a low interval, what they based this assessment on.
Comment 4 (Fifthly..): The reviewer understands that the authors designed the study to account for plot variation, but the importance of the variation still remains. If the variation within each fire interval was higher than the variation between them, then this will override many statistical methods. Having variation is not a complication, but clearly stating it and showing it helps the reader to properly account for it while it facilitates transparency of the findings and the study system. The authors can choose to ignore this comment if they want, but it lowers the overall quality of the manuscript.
Author Response
Please see the attachment.

This manuscript is a resubmission of an earlier submission. The following is a list of the peer review reports and author responses from that submission.
Round 1
Reviewer 1 Report
The paper is original and potentially important because it aims to evaluate the response of plant traits related to flammability in a tropical grassland ecosystem where little is known about this topic. It presents a good experimental design and enough field sampling, resulting in an interesting database on functional traits. However, I have found important aspects that need to be solved to make the manuscript publishable. These are detailed below:
Major comments:
-Scale of the study (line 48): Variables that determine flammability can be measured or operate at different scales, from organ to ecoregion (see Pausas et al., 2017); at the individual level they usually relate to traits such as branch architecture, dead tissue retention, etc; at the organ level, such as being leaf attributes like those measured in this work. This differentiation should be made clear beforehand (in the intro and in Fig 2) and emphasis should be placed on which scale is appropriate for each evaluated variable. This is important because the attributes at the organ level (smaller scale) can potentially modify the flammability of a plant, as there are other variables of higher hierarchy that can condition the overall response of a plant. For example, there may be variations in organ traits that cause the flammability of a plant to be modified when traits at the individual level (branching) also do so; or conversely, the flammability of a plant may vary when only individual traits (and not organ traits) do so.
That is, leaf level is a minor scale that can potentially affect flammability and scale to affect individual/population/community flammability, but they are not necessarily directly correlated.
-Adequacy of topics in the introduction: There are two paragraphs in which the conceptual contribution they make to the objective of the work is not understood. L 71-80: Although the wording is correct, it is not understood what is the relationship between the different types of plants (flammable/non-flammable) with the objective of the work. Will the information on traits of species with these different strategies be part of the objective and make a novel contribution to tropical savannas? The authors should explain what new contribution their study will make in relation to this conceptual framework. L 81-89: The relationship between different post-fire regeneration modes and flammability is not clear. Will flammability traits be analyzed for species with different regeneration types?
-Contextualization and hypotheses: A higher effort should be made to contextualize the object of study (functional groups/species), the environment and the scale of the study (intraspecific, interspecific, community) before developing hypotheses. The responses to the fire to be hypothesized may vary depending on these factors, e.g. life form. In addition, when reading the title, one may wonder why it is important to study non-graminoid species in grassland, where the most dominant species would be graminoids (grasses). According to what is stated in lines 130-131, they have studied the species with the highest abundance (which seems strange to me being in a grassland). However, from Table S1 it can be seen that most of the species studied are exclusive to only one fire frequency condition. The authors should make a greater effort to justify the selection of the group of plants studied.
Regarding hypothesis, the first one seems somewhat counter-intuitive and also goes against what is known about more frequent fires favouring short-lived, fast-growing herbaceous plants (grasses and even dicots) with high flammability (e.g. Pausas et al., 2017). Moreover, this hypothesis is somewhat circular: "lower flammability with increasing fire frequency due to persistence of less flammable species"; this is not very informative and does not say much about the mechanism. Why do the authors associate lower flammability with fire-tolerant species?
-Data analysis: The nature of the study design conditions the type of analysis. Nearby sites representing a fire frequency condition are not truly independent, nor are the measured plants within the same site. Thus, nearby sites of the same condition are more likely to have a more similar response to each other than to sites of a more distant condition. Therefore, by violating the lack of independence, a linear model (anova) analysis cannot be applied (e.g. see Zuur et al., 2009. Mixed Effects Models and Extensions in Ecology with R). In other words, the hierarchical structure of the data makes it necessary to use a model that takes into account these sources of variation (LMM). In this way, the authors will avoid calculating average values between plants and between sites as they did with their two-step method (lines 149-152) and the full power of the database will be exploited.
Also, the authors claim to have analyzed the data for each species, but then show trait results for all species together. I think that by showing trait data for more or less related species they are missing the information on how much variation the species identities are contributing to the general community pattern. So the recommendation would be to incorporate species identity as a random factor in the mixed models and see what proportion of the variance they explain.
-Results: With the current way in which the data were analyzed it is very difficult to distinguish whether the response patterns reported in the evaluated traits are due to variations in the fire regime or to site-specific variations. In fact, from Table S1 it is observed that of the 38 species only 5 are shared by at least two fire regimes and only 1 species is common in all three scenarios. However, in lines 166-168, this information is not reported. This could indicate that the possible community patterns found are driven by a change in the composition of species present. Compositional analyses (ANOSIM, NMDS) could provide evidence of changes in plant assemblages.
-The Discussion of the paper does not have much support considering that the data analyses are not adequate. I invite the authors to modify this section according to the suggested analyses and according to the scale of the evaluated traits (organ/plant).
Minor comments:
Lines 12-14: In the abstract, it is not clear what the working hypothesis is (why it is important to measure these traits in relation to fire?) and also whether the study is at inter- or intraspecific level or both.
L 35-36: It is not understood with respect to which variables or attributes the species or functional groups are affected by the fire regime. Specify whether it is with respect to their abundance or trait values.
L 43: Traits do not have their own entity but are attributes of the organisms. I suggest modifying the wording.
L 40. The authorship for this species' name is sketchy. However, I do not believe it is necessary to include the authors' taxonomic information on the species in the introduction to this ecological paper.
L 56-57: Explicit mention should be made here of "leaf economics"; what is the conceptual contribution of this term/framework to the work? It is mentioned in the title, but the introduction does not refer to the meaning of this term.
L 61-62: There are studies in the Chaco where an opposite pattern has been found (decrease of leaf morpho-physiological traits as fire frequency increases, Carbone & Aguilar 2017, Austral Ecol; or no effect of fire on traits of woody species, Santacruz-Garcia et al., 2019, Austral Ecol). It could be discussed at the end of the paper what the opposite responses are due to.
L 103: Is it included in the tropical savanna/grassland/shrublands biome? It would be opportune to make it explicit.
L 117-120: They should report the post-fire time at which the measurements were made. Do all regimes share the same post-fire time?
L 132: There were performed the measurements in all the non-graminoid species present in the plots, regardless of their abundance level? Was calculated the abundance of each species?
L 133: “branchiness”: Perez-Harguideguy et al. refer to this variable as "Branching architecture".
L 135-137: As mentioned before, this should be justified before, in the intro with the contextualization of the object of study. In addition, if graminoids present a continuous fuel layer through all different fire regimes, they should cite the reference. It could be asked if grasses do not present variations in their flammability, as there are precedents that show this for other spp (Bromus tectorum in the US).
L 142: As I mentioned before, they should group these traits according to whether they are traits at the level of the whole plant or at the level of organs (see Pérez-Harguindeguy et al., 2013).
L 142-143: To quantify these traits, how many organs (leaves/stems) per plant were measured? All technical specifications of how these variables were measured should be included, at least in a supplementary file.
L 154: “linear models (lm)” you mean instead of “linear fixed effect models”?
L 169-171: This should go in Methods.
Fig1. I suggest removing box b because if I understand correctly, the low-frequency sites are also shown in box c with a higher zoom. Also, I suggest adding a scale to the boxes where the sites are shown to know the distance that separates them.
Table 1: Does the data shown correspond to mean values and standard error or standard deviation?
Table S1: Would be more informative if instead of presence data for each species, you may report average abundance (if possible) or the number of plots in which the species were registered for each fire frequency condition.
Reviewer 2 Report
See attached

Reviewer 3 Report
This study investigated the plant traits in association with their flammability and leaf economics in a mountainous grassland in Brazil. Non-graminoid plant species were selected, and several plant features and parameters were measured, including leaf dry matter content, twig dry matter content, leaf area, bark thickness, branching, plant height and leaf toughness and specific leaf area. It was found that plant traits responded differently to the increase in fire frequency. This study somehow brings evidence that plant features respond coordinately towards a reduction in flammability with the increased fire frequency.
The results to support this study are shown in Figure 3 and Table 1. On one hand, the authors did not specify how to determine the fire frequency without taking into account the fuel moisture content; on the other hand, no efforts were made to demonstrate the reliability of the classified fire frequencies in the studied areas. Although the data analysis method was indicated by the authors, there is no direct evidence to exclude some key parameters in the assessment of the plant flammability, including moisture content of the fuels. At least, the authors need to elucidate why the omission of such a key parameter should be introduced in the present study.
A diversity of fire regimes was observed in the plant communities around the world. In terms of fire regimes, it seems inappropriate to specify which one is greater than the others. Nevertheless, we can say which plant stands or ranges own higher fire frequencies. The present study only evaluated the fire frequency parameter, as one of the parameters applied to define the fire regimes in a specific forest area. Attention should also be paid to the other components in the fire regime family, such as potential fire burning intensities and severity. In addition, please note that flammability is specific to certain materials, which is a terminology to define the fuel property such as ignitability, combustibility and so on, rather than a normal term used in the trial of the fire regimes in specific areas.
This paper is incomplete because of insufficient data exhibited in the section of Results and the missing concluding part. The way of writing is hard to follow. The paper at its current status is not suitable to be accepted by the journal.